# Oriented Vehicle Detection in Aerial Images Based on YOLOv4

**DOI:** 10.3390/s22218394

**Published:** 2022-11-01

**Authors:** Tai-Hung Lin, Chih-Wen Su

**Affiliations:** Department of Information & Computer Engineering, Chung Yuan Christian University, Taoyuan 320, Taiwan

**Keywords:** object detection, oriented bounding box, aerial image

## Abstract

CNN-based object detectors have achieved great success in recent years. The available detectors adopted horizontal bounding boxes to locate various objects. However, in some unique scenarios, objects such as buildings and vehicles in aerial images may be densely arranged and have apparent orientations. Therefore, some approaches extend the horizontal bounding box to the oriented bounding box to better extract objects, usually carried out by directly regressing the angle or corners. However, this suffers from the discontinuous boundary problem caused by angular periodicity or corner order. In this paper, we propose a simple but efficient oriented object detector based on YOLOv4 architecture. We regress the offset of an object’s front point instead of its angle or corners to avoid the above mentioned problems. In addition, we introduce the intersection over union (IoU) correction factor to make the training process more stable. The experimental results on two public datasets, DOTA and HRSC2016, demonstrate that the proposed method significantly outperforms other methods in terms of detection speed while maintaining high accuracy. In DOTA, our proposed method achieved the highest mAP for the classes with prominent front-side appearances, such as small vehicles, large vehicles, and ships. The highly efficient architecture of YOLOv4 increases more than 25% detection speed compared to the other approaches.

## 1. Introduction

With the rise of convolutional neural networks (CNNs) in the past decade, object detection performance has been rapidly enhanced based on its rich feature representation. Various detection schemes have been proposed to make good use of the features. They are mainly divided into two categories: two-stage detectors [1,2,3,4] and single-stage detectors [5,6]. Although each has its merits, they both use Horizontal Bounding Boxes (HBB) to locate all kinds of objects from arbitrary views. However, for some specific applications, such as vehicle/building detection in remote sensing images and scene text detection, an Oriented Bounding Box (OBB) fits the target more closely than an HBB. On the other hand, objects with large aspect ratios and that are densely arranged also make it challenging for detectors to locate the entire object area while excluding the non-object area by HBB. Previous research has shown that OBB can help detect the above cases, which usually occur in remote sensing images. Furthermore, the orientation of an object may also provide vital information for further processing.

Recently, increasingly oriented object detectors, which adjust based on the classic object detector, have achieved promising results in remote sensing. At the same time, the maturity of cost-effective drones in manufacturing has led to the widespread use of aerial photography in many areas, such as precision agriculture, rescue work, wildlife monitoring, etc. Among the numerous aerial photography applications, traffic surveying is one of the applications that requires accurate vehicle location via OBB rather than HBB. Since most vehicles can be adequately located by OBB in the bird view, the area of each vehicle can be efficiently represented by the corners of OBB. The estimation of OBB is usually accomplished by the regression of its representation model, which usually contains five to eight parameters. However, many previous researchers calculated these parameters based on the limited quadrilateral and OBB’s angle definitions in OpenCV. Ambiguous corner ordering and angular periodicity cause the discontinuous boundary problem and obstruct the regression of parameters [7,8]. The boundary discontinuity problem often causes the model’s loss value to suddenly increase at the boundary situation.

Aiming to solve the problem above, we propose a simple, practical framework for oriented object detection, which can be used in traffic surveys. To avoid the angular discontinuity, we regress the front point offset of the object rather than its angle.

As mentioned above, the HBBs have been widely used to indicate the position and area of objects in general object detection tasks. For instance, the classic two-stage object detector Faster RCNN [1], which evolved from RCNN [2] and Fast RCNN [3], is to filter out the proposals from Region Proposal Network layer (RPN) in the first stage and goes through a second stage for position tuning and objects classification. RPN requires pre-defined anchors to generate proposals also designed in the form of HBB. R-FCN [4] combined Region of Interest (ROI) pooling and position-sensitive score maps to improve detection speed and alleviate object position insensitivity in CNN. On the other hand, a one-stage detector, such as YOLO [5], SSD [6] or RetinaNet [9], evaluates images in a single phase to classify and locate objects simultaneously without second-stage calibration. Although the accuracy of a one-stage detector is lower than that of a two-stage detector initially, a single-stage structure allows for higher detection speeds. To bridge the gap inaccuracy, some functional mechanisms, including anchor box, Feature Pyramid Network (FPN) [10] and Cross Stage Partial Network (CSPNet) [11], are gradually introduced in the YOLO series [12,13,14,15]. The state-of-the-art object detectors mentioned above all use HBB to describe the position of arbitrary shape objects.

In order to obtain the position and orientation of rectangular objects while eliminating background interference, OBB becomes a better alternative to HBB. Researchers have proposed many fascinating oriented object detectors based on two-stage and one-stage models. Earlier oriented object detectors were based on the two-stage scheme and applied to detect scene text. RRPN [16] directly adds an angle parameter to the RPN of Faster RCNN to make proposals directional. Then, the arbitrarily oriented proposals were converted to the fixed-size feature maps by Rotation Region of Interest (RROI) pooling to detect the oriented scene text. In $R^{2}\mathrm{CNN}$ [17], Jiang Y. et al. mentioned that the shape of a bounding box is very similar, but the angle difference is enormous when rotated $90^{{}^{\circ}}$ and $-90^{{}^{\circ}}$. The discontinuity of the angle may cause difficulties in network learning. Therefore, they adopted the coordinates of the first two corners clockwise and the height of the bounding box to represent an inclined rectangle (*x1, y1, x2, y2, h*). To catch significant aspect ratio texts, they also performed ROI pooling with different pool sizes (7 × 7, 11 × 3, 3 × 11). To compensate for the lack of angle information in the general object detector, the ROI transformer [18] learns the transformation parameters to convert horizontal ROI to the rotated ROI and extract rotation-invariant features for the classification and localization afterward. In recent years, more and more oriented object detectors have been proposed and applied to aerial photographs. Due to the large variability of object size/orientation and the complexity of the background in aerial scenes, many orientated object detectors focus on deeper feature extraction and attention mechanisms. R-DFPN [19] proposed a dense feature pyramid network (DFPN) to better integrate features of different scales to increase the usage rate of features. ICN [20] combines multiple modules, such as an image cascade network, feature pyramid network and deformable inception network, to achieve satisfactory performance in detecting oriented objects. SCRDet [7], CADet [21] and RADet [22] add different attention modules before RPN to better handle small objects and cluttered oriented objects. A gliding vertex [23] guarantees detection accuracy through regressing vertices offset without changing network architecture.

Due to the lower computational cost and memory requirement, some oriented object detectors were developed based on a one-stage scheme. Textbox++ [24] and RRD [25] adopt the vertex regression on SSD while the latter changes the backbone convolution into oriented response convolution. Considering the difficulty of angle training in R-DFPN, SAR [26] converts some of the rotating parameters from the angular regression task to the classification task to avoid boundary and order problems. CSL [27] uses the circular smooth label technique to obtain a more robust angular prediction through classification without suffering boundary conditions caused by periodic angles. Unlike a two-stage detector, a single-stage detector does not have ROI pooling or an ROI align phase to reduce the impact of feature misalignment. $R^{3}\mathrm{Det}$ [28] adds a refinement stage based on RetinaNet to achieve feature alignment by reconstructing the feature map and using a deeper network backbone to extract richer features. In addition to adding the refinement stage, RSDet [8] directly regresses the four corner coordinates of the oriented bounding box to avoid inherent regression inconsistencies. A modulated rotation loss is designed to address the problem of loss discontinuity.

No matter two-stage or one-stage oriented object detectors, there are three main ways to determine an oriented bounding box: the five-parameter regression-based method, he eight-parameter regression-based method and the classification-based method. Five-parameter regression-based methods usually introduce an additional parameter θ to regress the angle of the bounding box. The range of θ is usually limited to 90 degrees [7,19,21,28] or 180 degrees [16]. However, the network is prone to suffering from the loss discontinuity problem in the boundary case. In eight-parameter regression-based methods [8,20,23,24,25], the coordinates of the quadrilateral’s four corners are regressed. The boundary discontinuity problem still exists due to the ordering of corner points. Classification-based methods [27,29] turn the angle regression task into the classification task, which truly addresses the essence of boundary problem. Nevertheless, it loses the precision caused by angle discretization and also increases the difficulty of tuning the network caused by importing the new hyperparameter window function. Moreover, almost none of the above methods can directly determine the front side of an object. It is practical to provide the front information of an object in traffic analyses or sailing analyses. 

To achieve highly efficient oriented object detection, we adopt YOLOv4 architecture and extend its head part to estimate object orientation. To overcome the boundary discontinuity problem, the front point of the object is introduced. Experimental results demonstrate that our proposed method outperforms the others in terms of vehicle detection, which plays a vital role in traffic analysis.

## 2. Materials and Methods

### 2.1. The Problems of Regression-Based Rotating Object Detector

As described in Section 1, the Boundary Discontinuity Problem (BDP) is a significant obstruction of the estimation of OBB. An OBB can be determined by the center (x,y), width w, height h and angle θ of it, or just by its four corners $\left(x_{1},y_{1}\right)$, $\left(x_{2},y_{2}\right)$, $\left(x_{3},y_{3}\right)$, and $\left(x_{4},y_{4}\right)$. For the former five-parameter OBB models, the angle θ is defined as the acute angle between the x-axis and the first side (width) of OBB that it touches counterclockwise as the definition in OpenCV, limited to the range [−π/2, 0) as the Figure 1a. In this case, the acute angle could be formed by x-axis and any side of OBB, which means the width w and height h could easily be confused and cause Exchangeability of Edges (EoE). To alleviate the impact of EoE, some studies further define θ as the acute angle between the x-axis and the first long side of OBB, thus extending the range from [−π/2, 0) to [−π, 0). The length of the long side is then explicitly specified as the width w and that of the short side is specified as the height h as shown in Figure 1b. However, the value of the acute angle is still not continuous, while the long side of OBB is almost parallel to the x-axis. The above periodicity of angular (PoA) may cause significant loss changes once the angle falls at either end of its permissible range.

Compared to the five-parameter OBB model, which introduces an additional θ parameter to indicate the orientation of the object, the eight-parameter OBB model discards the angle information. Although the problem of PoA does not occur in the regression of OBB’s four corners, the consistency of the alignment of these four corners remains an issue. In addition, the eight-parameter OBB model determines rectangles and arbitrary quadrilaterals, which may be undesirable when representing artificial buildings and vehicles.

### 2.2. Front Point Offset Regression for Angle Prediction

To overcome the EoE and PoA problems caused by the five-parameter OBB model and to extend the angle range to [−2π, 0), the parameter θ is replaced by a couple of parameters $x_{f}$ and $y_{f}$, where ($x_{f}$, $y_{f}$) indicates the midpoint of the front side of the objects, such as vehicles and buildings. In this case, an OBB is represented by six parameters, as shown in Figure 1c, and width is defined as the length of the front/rear side and height refers to the length of the other side. Inspired by the YOLO grid cell strategy, the angle prediction is converted to that of the front point offset $\left(b_{xf},b_{yf}\right)$ with respect to the grid, as shown in Figure 2. Network outputs $\left(t_{x},t_{y},t_{w},t_{h}\right)$ pass through a sigmoid function σ and adjust the size with the predefined anchor $\left(p_{w},p_{h}\right)$ to reconstruct the predict bounding box ($b_{x},\;b_{y},\;b_{w},\;b_{h},\;b_{xf},\;b_{yf}$). Here, the front point and center respect the same grid for each OBB to make the training more convergent. Since the prediction of angle is translated from periodic angle space into the distance space domain, the PoA problem no longer exists. In addition, the assignment of the front point makes the alignment of corners no longer ambiguous and confusing, which fundamentally avoids the EoE problem.
(1)$$b_{x}=2\sigma \left(t_{x}\right)-0.5+C_{x}$$
(2)$$b_{y}=2\sigma \left(t_{y}\right)-0.5+C_{y}\;b_{w}=p_{w}{\left(2\sigma \left(t_{w}\right)\right)}^{2}$$
(3)$$b_{h}=p_{h}{\left(2\sigma \left(t_{h}\right)\right)}^{2}$$
(4)$$b_{xf}=t_{xf}+C_{x}$$
(5)$$b_{yf}=t_{yf}+C_{y}$$


### 2.3. Overview of the Network

The proposed network architecture is based on YOLOv4, as illustrated in Figure 3. YOLOv4 is one of the fastest and most accurate object detectors available today. Its architecture consists of three parts: backbone, neck and head. The backbone, CSPDarknet53, in YOLOv4 optimizes darknet53 with Cross Stage Partial Network (CSP) to reduce the reuse of gradient information, thereby decreasing computational cost and increasing learning ability. Yolov4 utilizes Spatial Pyramid Pooling (SPP) for expanding the field of view in the neck part. It combines Feature Pyramid Networks (FPN) and Path Aggregation Networks (PAN) for fusing low-level and high-level features from different scales to achieve better localization. The head part is the prediction layer which contains a certain amount of grid cells at three different scale levels. Each grid cell outputs the prediction results into a (C+5)×A dimensional vector, with C representing the probability of each class, and A is the number of predefined anchors in each prediction layer. The remaining five dimensions encode a horizontal bounding box into x, y, w, h and confidence. We take advantage of the above YOLOv4 architecture and add two additional parameters $t_{xf}$ and $t_{yf}$ for each grid cell to predict front point offset. As a result, the proposed method can localize objects by OBB and obtain the following advantages:1.Avoid boundary discontinuity problem

Both the angle regression and corner regression methods will produce certain boundary discontinuity problems that lead to instability in loss calculation. In contrast, our proposed method regresses front point offset and avoids the sharp increase of loss generated by the boundary discontinuity problems. 

2.Less predefined anchors

It is not only necessary to use anchors of different sizes and aspect ratios to provide anchors closer to the ground truth for the five-parameter oriented object detector but also to add anchors at multiple angles to help the model converge, which also increase computational cost in the process. Our approach does not focus on the regression of angle. We regress the front point offset to maintain the overall performance of the model without increasing the number of anchors.

3.Explicit front end information

Since our method locates the front point coordinates of the object, we can avoid boundary discontinuous and obtain the object’s orientation within [−2π, 0). This helps to indicate the front end of the stopped vehicle and to understand whether the vehicle is in violation and the possible direction of travel. In contrast, the other OBB definitions are more likely to face the discontinuity problem of extreme values of angles and provide limited angle ranges.

### 2.4. Loss Function

The loss function is decomposed into four parts, which are defined as follows:(6)$$L=L_{cls}+L_{obj}+L_{box}+L_{fp}$$
(7)$$L_{cls}=\frac{\lambda_{1}}{N_{pos}}\overset{S^{2}}{\underset{i=0}{\sum }}\overset{A}{\underset{j=0}{\sum }}1_{ij}^{obj}\underset{c\epsilon Classes}{\sum }[{\hat{p}}_{ij}\left(c\right)\log\left(p_{ij}\left(c\right)\right)+\left(1-{\hat{p}}_{ij}\left(c\right)\right)\log\left(1\;-p_{ij}\left(c\right)\right)]$$
$$L_{obj}=\frac{\lambda_{2}}{N}\overset{S^{2}}{\underset{i=0}{\sum }}\overset{A}{\underset{j=0}{\sum }}1_{ij}^{obj}\left[{\hat{C}}_{ij}\log\left(C_{ij}\right)+\left(1-{\hat{C}}_{ij}\right)\log\left(1-C_{ij}\right)\right]+$$
(8)$$1_{ij}^{noobj}\left[{\hat{C}}_{ij}\log\left(C_{ij}\right)+\left(1-{\hat{C}}_{ij}\right)\log\left(1-C_{ij}\right)\right]$$
(9)$$L_{box}=\frac{\lambda_{3}}{N_{pos}}\overset{S^{2}}{\underset{i=0}{\sum }}\overset{A}{\underset{j=0}{\sum }}1_{ij}^{obj}L_{IoU}$$
(10)$$L_{fp}=\frac{\lambda_{4}}{N_{pos}}\overset{S^{2}}{\underset{i=0}{\sum }}\overset{A}{\underset{j=0}{\sum }}1_{ij}^{obj}smooth_{L1}\left(\hat{fp},fp\right)$$
where $S^{2}$ is the number of grid cells in the feature map and A denotes the number of anchors in each grid cell. $N_{pos}$ is the number of positive samples. *N* is the number of positive samples and negative samples. $L_{cls}$ denotes classification loss. $L_{obj}$ denotes objectness loss. We use binary cross entropy to calculate both $L_{cls}$ and $L_{obj}$. $L_{box}$ is the OBB regression loss based on IoU (Intersection over Union). $L_{fp}$ denotes the regression loss of front point offset. $1_{ij}^{obj}$ and $1_{ij}^{noobj}$ are both binary values, which indicate positive samples or not. $\lambda_{1}$ to $\lambda_{4}$ are the hyper-parameters to balance the weights of each loss term.

Since the factor of the direction of front point is not considered in the IoU, the IoU can be very small, even if the front point of two OBBs is located in a completely different direction. To penalize the loss of IoU by using the angular difference caused by the front points, the correction of the vectors formed by the center point and front point is introduced in $L_{IoU}$ as below:(11)$$L_{IoU}=IoU\cdot \cos\left(\frac{∠\left(\vec{P_{c}P_{f}},\vec{T_{c}T_{f}}\right)}{2}\right)$$
where $P_{c}$ and $P_{f}$ indicate the center point and front point of a predicted box. $T_{c}$ and $T_{f}$ represent the center point and front point of ground truth box. The cosine value of half of the angle formed by vectors $\vec{P_{c}P_{f}}$ and $\vec{T_{c}T_{f}}$ is in the range [0, 1], which ensures that the value of $L_{IoU}$ remains in the range [0, 1].

## 3. Datasets and Results

The proposed method was implemented by PyTorch [30] with Tesla V100 and 32G memory. To verify the performance of our proposed method, we conducted experiments on three datasets, which contain OBB labeling information.

### 3.1. Datasets and Evaluation Protocols

#### 3.1.1. DOTA

DOTA [31] is one of the large-scale datasets for object detection in the aerial image. It contains 2806 images from different sensors and platforms. The size of images range from approximately 800 × 800 to 4000 × 4000. The fully annotated DOTA images comprise 188,282 instances, each of which is annotated by an arbitrary (8 d.o.f) quadrilateral in 15 classes: plane (PL), baseball diamond (BD), bridge (BR), ground track field (GTF), small vehicle (SV), large vehicle (LV), ship (SH), tennis court (TC), basketball court (BC), storage tank (ST), soccer ball field (SBF), round about (RA), harbor (HA), swimming pool (SP) and helicopter (HC). Half of the original images were randomly selected as the training set, 1/6 as the validation set, and 1/3 as the testing set. We cropped original images into 1024 × 1024 patches with a stride set to 512. Finally, we obtained 29,457 patches for the training process.

#### 3.1.2. HRSC2016

HRSC2016 [32] is a high-resolution ship dataset collected from six different harbors on Google Earth. It contains 1061 images with sizes ranging from 300 × 300 to 1500 × 900. A rotating rectangle labels ground truths. We followed the original division to obtain the training, validation and test sets with 436, 181 and 444 images, respectively. The size of HRSC2016 is significantly smaller than the DOTA in terms of the number of images, the number of categories and the diversity of content. In DOTA1.0, the number of ships used for training was around tens of thousands. Most of the images contain dozens to thousands of targets. On the other hand, HRSC2016 has about 3000 targets in all.

#### 3.1.3. Our Dataset

To verify the robustness of the proposed method in traffic applications, we used UAVs to capture images from several road intersections at 65~100 m above the ground. All the images are sized 1920 × 1080. Unlike remote sensing images, detection targets and vehicles are much closer to the camera, making the targets look much more significant in our images. We randomly selected 723 images for training and validation and 180 for testing. All images are annotated with OBB in eight classes: sedan, truck, bus, tractor, trailer and motorbike. The number of each class is listed in Table 1.

### 3.2. Training Details

CSPDarknet53 is the backbone of architecture, which was pretrained on COCO [33] and provided by YOLOv4. The stochastic gradient descent (SGD) was used as an optimizer in all datasets, and its weight decay and momentum were set to 0.0005 and 0.937, respectively. Training epochs and learning rates were set differently in each dataset. We trained 250 epochs on DOTA and 300 epochs on HRSC2016 and our dataset. The initial learning rates for DOTA, HRSC2016 and our dataset were 0.01, 0.0001 and 0.05, respectively. The mini-batch sizes of DOTA and HRSC2016 were set to eight and set to one for our dataset. The augmentation method is applied to all datasets, including mosaic, random flip, random color, random rotation and random scaling.

### 3.3. Result on DOTA1.0

Table 2 shows the comparison results on DOTA. The results here are obtained from the official DOTA evaluation server. Our proposed method attains a good trade-off between accuracy and speed. With test time augmentation (TTA), the mAP of our proposed method achieves 73.89% with 20.4 fps on average. Our results outperform all the other methods at the same speed level. Moreover, the three categories: small vehicle, large vehicle and ship obtain the highest mAP 79.37%, 83.34% and 88.65% in all methods. This may be due to the obvious orientation of these three types of objects and the fact that the bounding boxes of the objects are more rectangular in shape. Among the methods that are better than ours in the table, Gliding Vertex [23], Mask OBB [34], FFA [35], APE [36] and CenterMap OBB [37] are based on the slower two-stage detectors and RSDet [8], GCL [29] and CSL [27] are based on the single-stage methods while using deeper backbone (ResNet152). In terms of speed, our method is 5 fps faster than SAR [26], which is the fastest method on record in Table 2. The first reason is that our proposed approach is based on the YOLOv4 architecture, which is designed for high efficiency. Second, in some oriented object detectors, many anchors with different sizes and angles are proposed for better detection results. This leads to a significant increase in the number of detection candidates and increases the computation time when non-maximal suppression is applied. Our proposed six-parameter OBB model has the potential to maintain accuracy while keeping the minimum number of anchors.

### 3.4. Result on HRSC2016

HRSC2016 contains a large number of ships with arbitrary rotation and large aspect ratios, which is a great challenge to locate accurately. The comparison results are listed in Table 3. With TTA, the mAP of our proposed method achieves 93.7% under voc2012 evaluation protocol. It is worth noting that the inference speed for most methods is around 10 fps and that of the fastest method is 15.53 fps for 800 × 800 images. In comparison, our method can reach an inference speed of 19.23 fps and even 52.63 fps without TTA, which is more than five times faster than most SOTA methods.

### 3.5. Result on Our Dataset

Our dataset evaluation results and confusion matrix are shown in Table 1 and Figure 4. Vehicles are roughly divided into six categories: sedan, truck, bus, tractor, trailer and motorbike. Most of the large vehicles have an area of more than 4000 pixels, and the trailer may exceed 17,000 pixels. On the other hand, the area of a motorbike is less than 800. The mAP of a tractor and trailer only achieve 87.9% and 80.3% due to the lack of corresponding training samples. Nevertheless, the other three types of large objects can achieve over 93%, and sedans can even reach 98.1% mAP. In terms of small vehicles, the mAP of the motorbike is 80%. Low mAP is caused by two main reasons. The first reason is the small appearance of the motorbike in the aerial image. The crowded and irregular groups of motorcycles further increase the confusion on the feature map. Therefore, a motorbike could be easily mis-detected in a waiting zone or under the shade of trees. The second reason is that many incomplete motorbikes in the training images were mislabeled, which caused a high false positive rate, as shown in Figure 4.

Figure 5, Figure 6 and Figure 7 show some detection results on DOTA1.0, HRSC2016 and our dataset. The color of its OBB indicated the class of each detected object, and the white side denotes the front side of the object. As we can see, our model has outstanding detection results under various scales of aerial scenarios.

## 4. Discussion

The evaluation from the experimental results shows that our method has a good trade-off between accuracy and speed, especially when the object has an explicit front-side appearance. However, the proposed method has weaknesses in the following cases: (1) When an object does not have a visible front side, determining its front side could be ambiguous, as shown in Figure 8. (2) It is still difficult to precisely locate objects with very large aspect ratio, as shown in Figure 9. In the future, we will try to add a lightweight attention module to locate the front point or use deformable convolution to detect objects with large aspect ratios.

## 5. Conclusions

In this paper, we proposed an arbitrary oriented object detector based on the YOLOv4 framework. We introduced the front point offset of an OBB to avoid the discontinuous boundary problem caused by the regression of angle or corners. The proposed method is especially suitable for vehicle localization in aerial images. It not only generates the OBB fit for the vehicle but also indicates the front end of it, which plays a vital role in traffic analysis. With the validation of the public datasets DOTA and HRSC2016, our detector was able to reach promising detection accuracy and achieve outstanding inference speed. The inference speed outperforms all the other methods and can accomplish real-time detection. Since our approach focuses on changing the head area’s structure, it is easier to further improve the accuracy and speed with future upgrades of the YOLO series.

## Figures and Tables

**Figure 1 sensors-22-08394-f001:**
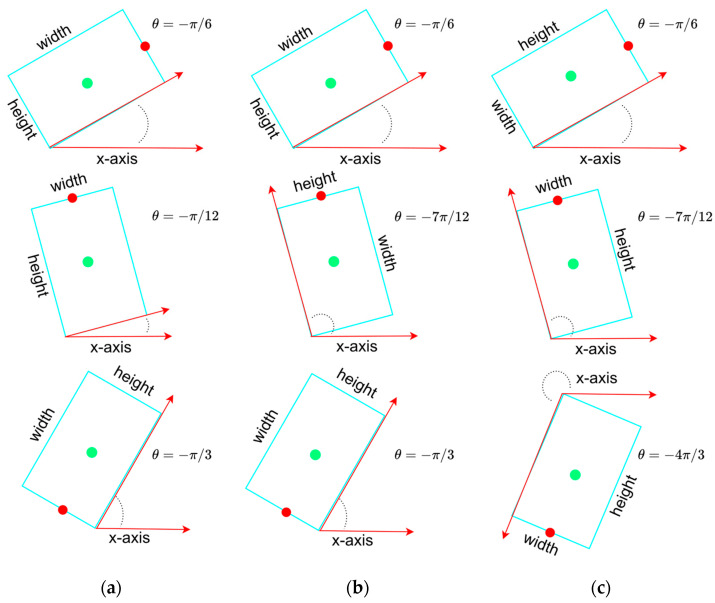
Three different definitions of the angle, width and heigh of the OBB. Green point and red point represent center point and front point of the OBB, respectively. (**a**) The definition of an OBB in OpenCV. θ ranges from −π/2 to 0. (**b**) The extended version of the OBB definition in OpenCV. In this case, the width of OBB is always set to the longer side. θ range from −π to 0. (**c**) The OBB definition in our proposed work. A front point is added to guide the angle prediction. θ ranges from −2π to 0.

**Figure 2 sensors-22-08394-f002:**
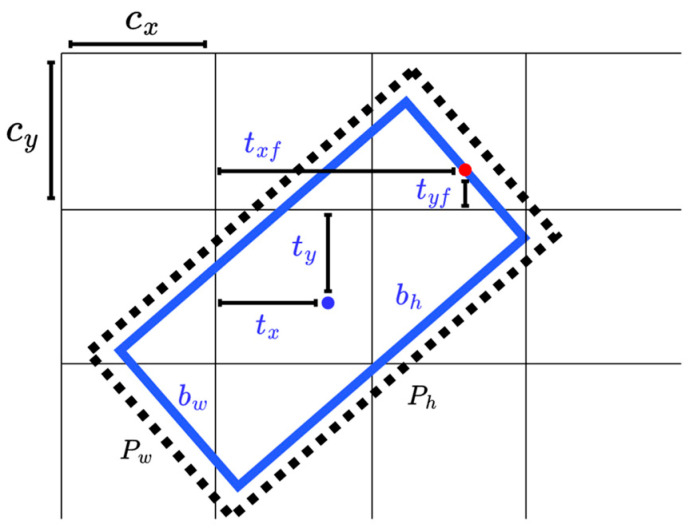
The network predicts 6 coordinates *t_x_*, *t_y_*, *t_w_*, *t_h_*, *t_xf_* and *t_yf_* of each bounding box. (*C_x_*, *C_y_*) is the cell offset from the top left corner of the image. Red point and blue point denote the front point and center point of bounding box, respectively. Both front point and center point share the same grid cell in the network.

**Figure 3 sensors-22-08394-f003:**
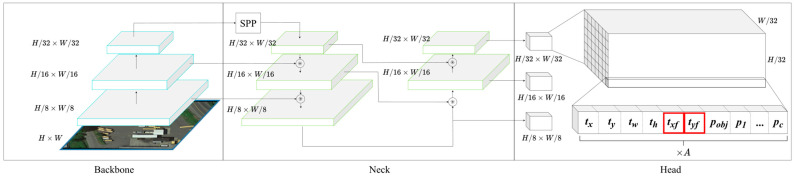
The network architecture of our proposed method. The feature extraction (backbone) and feature fusion (neck) parts are kept the same as YOLOv4. Meanwhile, two additional channels *t_xf_* and *t_yf_* are added into the grid cells (head) at different scale levels. Thus, each grid cell contains an (C + 7) A dimensional vector, where C is the number of classes, A is the number of predefined anchors in each prediction layer and each anchor contains seven parameters: *t_x_*, *t_y_*, *t_xf_*, *t_yf_*, *t_w_*, *t_h_* and confidence *p_obj_*. ⊛ denotes feature map concatenation.

**Figure 4 sensors-22-08394-f004:**
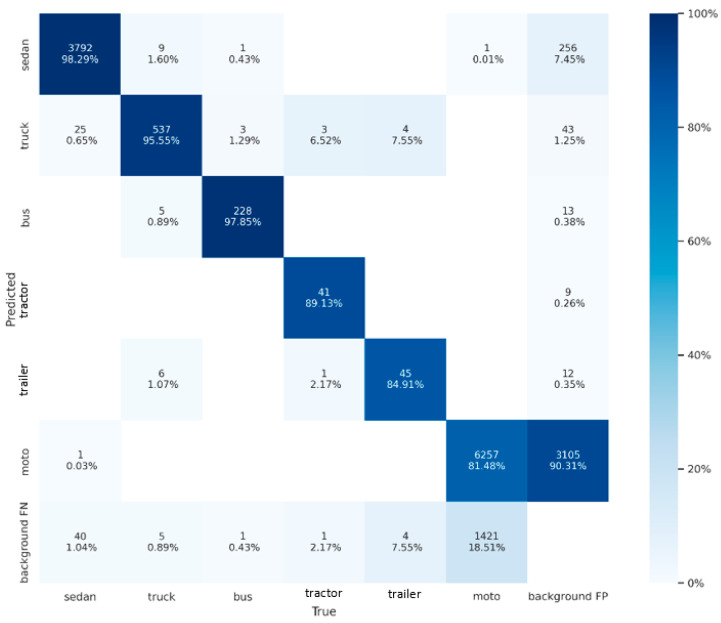
Confusion matrix of our dataset. The horizontal axis is the ground truth and the vertical axis is the predicted result. Confidence threshold and IoU threshold are set to 0.01 and 0.1, respectively.

**Figure 5 sensors-22-08394-f005:**
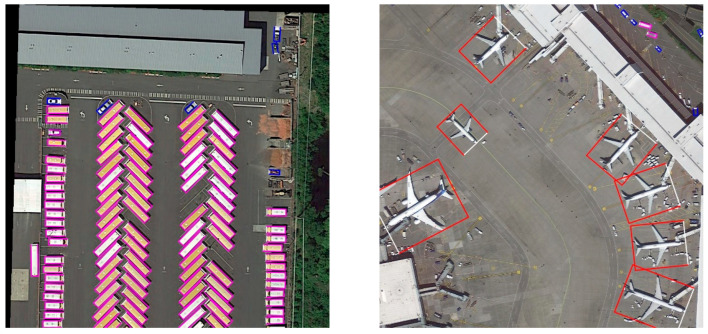
Detection results on DOTA1.0. The colors of OBBs are defined as follows: red: plane, yellow: baseball diamond, green: bridge, cyan: ground track field, blue: small vehicle, magenta: large vehicle, dark orange: ship, green yellow: tennis court, spring green: basketball court, deep sky blue: storage tank, dark violet: soccer ball field, deep pink: round about, orange: harbor, chartreuse: helicopter.

**Figure 6 sensors-22-08394-f006:**
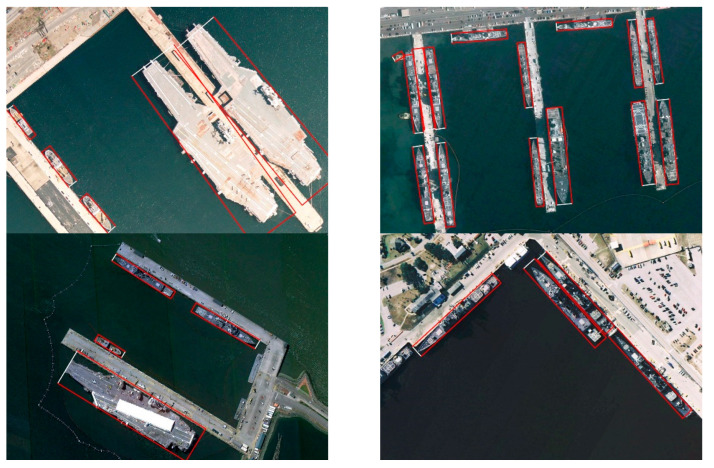
Detection results on HRSC2016.

**Figure 7 sensors-22-08394-f007:**
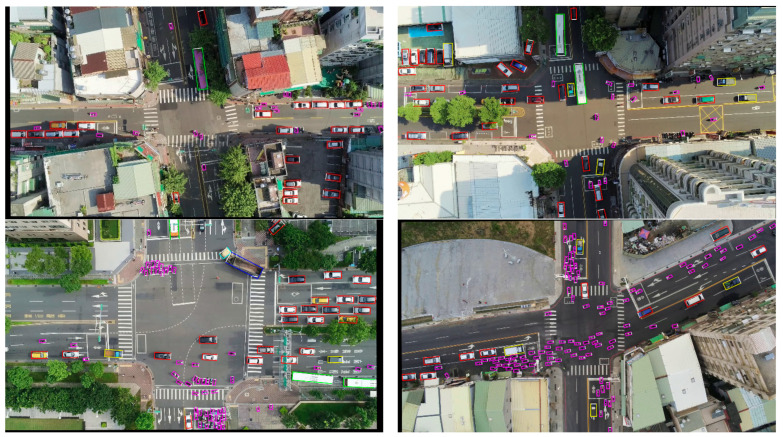
Detection results on our dataset. Bounding box color is defined as follows: red: sedan, yellow: truck, green: bus, cyan: tractor, blue: trailer, magenta: motorbike.

**Figure 8 sensors-22-08394-f008:**
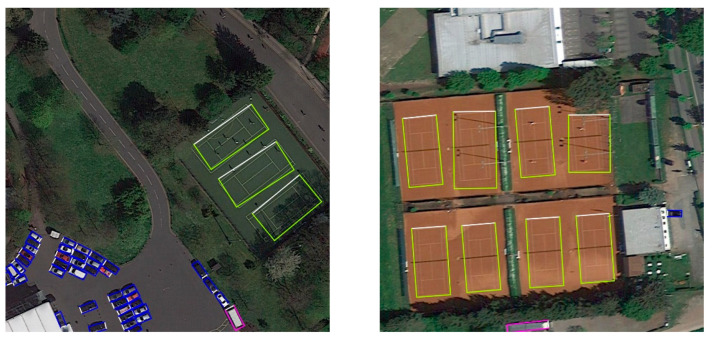
The ambiguous problem of determining the front side of the tennis court. Since any opposite side of a tennis court is symmetrical, it is hard to define its front side clearly. The left detection result treats the longer side as the front side while the right one treats the shorter side as the front side. This makes the detection results inconsistent and leads to imprecise OBB positioning.

**Figure 9 sensors-22-08394-f009:**
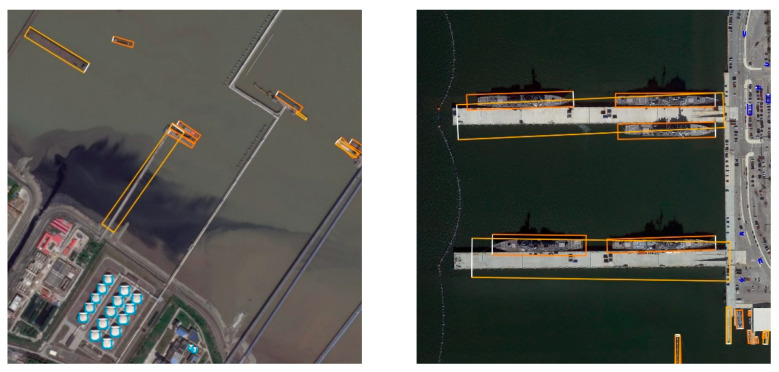
The OBB and front point of the object with very large aspect ratio is difficult to precisely locate.

**Table 1 sensors-22-08394-t001:** Evaluation results of our dataset.

Class	Ground Truth Counts	Precision	Recall	mAP
Sedan	14,924	0.978	0.975	0.981
Truck	2035	0.931	0.939	0.933
Bus	605	0.958	0.976	0.974
Tractor	153	0.938	0.891	0.879
Trailer	151	0.833	0.849	0.803
Motorbike	26,314	0.940	0.761	0.800
All	44,182	0.930	0.898	0.895

**Table 2 sensors-22-08394-t002:** The comparison results on the DOTA1.0 dataset.

Method	backbone	PL	SV	LV	SH	HC	BD	BR	GTF	TC	BC	ST	SBF	RA	HA	SP	mAP	FPS
$R^{2}\mathrm{CNN}\;$^(*II*)(*A*)^ [17]	ResNet101	80.94	59.92	50.91	55.81	48.22	65.67	35.34	67.44	90.67	66.92	72.39	55.06	52.23	55.14	53.35	60.67	
RRPN ^(*II*)(*A*)^ [16]	ResNet101	88.52	51.85	56.19	57.25	53.58	71.20	31.66	59.30	90.81	72.84	67.38	56.69	52.84	53.08	51.94	61.01	
RoI-Transformer ^(*II*)(*A*)^ [18]	ResNet101	88.64	68.81	73.68	83.59	47.67	78.52	43.44	75.92	90.74	77.27	81.46	58.39	53.54	62.83	58.93	69.56	5.90
SCRDet ^(*II*)(*A*)^ [7]	ResNet101	89.98	68.36	60.32	72.41	65.21	80.65	52.09	68.36	90.85	87.94	86.86	65.02	66.68	66.25	68.24	72.61	
MFIAR-Net ^(*II*)(*A*)^ [38]	ResNet152	89.62	70.13	67.64	77.81	62.11	84.03	52.41	70.30	90.85	85.40	86.22	63.21	64.14	68.31	70.21	73.49	
Gliding Vertex ^(*II*)(*A*)^ [23]	ResNet101	89.64	73.01	73.14	86.82	57.32	85.00	52.26	77.34	90.74	79.02	86.81	59.55	**70.91**	72.94	70.86	75.02	10.00
Mask OBB ^(*II*)(*A*)^ [34]	ResNeXt101 [39]	89.56	76.52	74.16	85.63	63.32	**85.95**	54.21	72.90	89.85	83.81	86.48	54.89	69.64	73.94	69.06	75.33	6.20
FFA ^(*II*)(*A*)^ [35]	ResNet101	90.10	71.00	79.90	83.50	63.70	82.70	54.20	75.20	90.70	83.90	84.60	61.20	68.00	70.70	76.00	75.70	
APE ^(*II*)^ [36]	ResNeXt101	89.96	74.01	77.16	79.45	65.55	83.62	53.42	76.03	90.83	87.15	84.51	**67.72**	60.33	**74.61**	71.84	75.75	
CenterMap OBB ^(*II*)(*A*)^ [37]	ResNet101	89.83	77.66	78.32	87.19	66.06	84.41	54.60	70.25	90.66	84.89	85.27	56.46	69.23	74.13	71.56	76.03	6.30
Cascade-FF [40]	ResNet152	89.90	68.20	75.20	75.60	50.10	80.40	51.70	**77.40**	90.80	78.80	84.40	62.30	64.60	57.70	69.40	71.80	
BBAVectors [41]	ResNet101	88.35	78.43	78.98	87.94	55.70	79.96	50.69	62.18	90.85	83.58	84.35	54.13	60.24	65.22	64.28	72.32	
SARD [42]	ResNet101	89.93	68.41	61.18	66.00	68.03	84.11	54.19	72.04	90.82	87.79	86.59	65.65	64.04	66.68	68.84	72.95	
GLS-Net [43]	ResNet101	88.65	73.30	72.16	84.68	60.42	77.40	51.20	71.03	90.87	80.43	85.38	58.33	62.27	67.58	70.69	72.96	
DRN [44]	Hourglass104 [45]	89.71	76.22	74.43	85.84	58.48	82.34	47.22	64.10	90.57	86.18	84.89	57.65	61.93	69.30	69.63	73.23	
FADet [46]	ResNet101	90.21	73.18	68.27	79.56	64.86	79.58	45.49	76.41	90.83	83.40	84.68	53.40	65.42	74.17	69.69	73.28	
SAR [26]	ResNet101	**90.89**	68.07	72.31	81.24	63.48	82.67	49.75	69.90	90.27	**88.19**	82.43	62.08	66.43	66.20	68.34	73.48	15.53
$R^{3}\mathrm{Det}\;$^(*A*)^ [28]	ResNet152	89.49	70.92	78.66	78.21	67.17	81.17	50.53	66.10	90.81	85.26	84.23	61.81	63.77	68.16	69.83	73.74	
RSDet ^(*A*)^ [8]	ResNet152	90.10	70.20	78.70	73.6	63.70	82.00	53.80	68.50	**91.20**	87.10	84.70	64.30	68.20	66.10	69.30	74.10	
GCL (FPN based) ^(*A*)^ [29]	ResNet152	89.70	78.98	74.78	85.86	68.91	83.34	**55.44**	67.31	90.82	85.56	85.33	65.56	61.52	72.30	**78.11**	76.23	
CSL(FPN based) ^(*A*)^ [27]	ResNet152	90.13	77.32	72.98	85.94	**70.08**	84.43	54.57	68.13	90.74	85.95	86.36	63.42	65.82	74.06	73.67	**76.24**	
Ours ^(*A*)^	CSPDarknet53	87.63	78.66	77.99	**88.68**	59.20	80.60	43.65	56.12	90.85	78.13	86.78	44.32	59.34	68.13	71.50	71.44	**35.70**
Ours ^(*A*)^ *	CSPDarknet53	88.94	**79.37**	**83.84**	88.65	59.78	82.19	43.22	60.15	90.83	83.52	**87.37**	51.30	66.27	72.52	70.46	73.89	20.40

(*A*) indicates that the method is anchor-based. (*II*) indicates that the method is a two-stage detector. *** indicates inference with test time augmentation. Red color used to highlight best values.

**Table 3 sensors-22-08394-t003:** The comparison results on HRSC2016.

Method	Backbone	Image Size	mAP (07)	mAP (12)	FPS
$R^{2}\mathrm{CNN}\;$[17]	ResNet101	800 × 800	73.07	79.73	2.0
RC1 & RC2 [47]	VGG16	800 × 800	75.70		<1.0
RRPN [16]	ResNet101	800 × 800	79.08	85.64	3.5
R^2^PN [48]	VGG16		79.60		<1.0
RetinaNet-H [28]	ResNet101	800 × 800	82.89	89.27	14.0
RRD [25]	VGG16	384 × 384	84.30		slow
RoI-Transformer [18]	ResNet101	512 × 800	86.20		6.0
RSDet [8]	ResNet152		86.50		
Gliding Vertex [23]	ResNet101		88.20		
SAR [26]	ResNet101	896 × 896	88.45		15.5
BBAVectors [41]	ResNet101	608 × 608	88.60		11.7
DRN [44]	Hourglass104			92.70	
CenterMap OBB [37]	ResNet50			92.80	
SBD [49]	ResNet50			93.70	
RetinaNet-R [28]	ResNet101	800 × 800	89.18	95.21	10.0
$R^{3}\mathrm{Det}\;$[28]	ResNet152	800 × 800	89.33	96.01	10.0
FPN-GCL-based [29]	ResNet101		89.56	96.02	
FPN-CSL-based [27]	ResNet101		89.62	96.10	
Ours	CSPDark53			90.40	52.6
Ours *	CSPDark53			93.70	19.2

* indicates inference with test time augmentation.

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
