# Peer review of "Oriented Vehicle Detection in Aerial Images Based on YOLOv4"

_sensors, 2022, doi:10.3390/s22218394_

Round 1

Reviewer 1 Report

l.26 [1][4] and more <- [1,4]

l.41 "..." to remove - there is "etc."

Eq.6 too small brackets
Eq.6 \cdot instead cross multiplication

Fig.2 - move formulas from image to formulas in the paper text

Table.1 and more
missing zero, e.g.: 87.8 <- 87.80

l.351 missing coma:
"DOTA1.0 ` HRCS2016"

Section Result - should be modified to the section Results and section Discussion

Negative results should be discussed

Author Response

Many thanks for your careful review. We have cautiously corrected typos in the manuscript and have also added a discussion section after the results section. Below is the point-by-point response to the reviewer’s comments.

Comment 1: l.26 [1][4] and more <- [1,4]

Response: We have fixed the error (line 28, 29).

Comment 2: l.41 "..." to remove - there is "etc."

Response: We have fixed the typo (line 42).

Comment 3: Eq.6 too small brackets

Eq.6 \cdot instead cross multiplication

Response: We have fixed the typo of this equation (line 253).

Comment 4: Fig.2 - move formulas from image to formulas in the paper text

Response: The formulas are isolated from the image (on line 185).

Comment 5: Table.1 and more

missing zero, e.g.: 87.8 <- 87.80

Response: We have padded zeros to two decimal places to all numbers in table 1, table 2 and table 3 (on line 303, 351, 353).

Comment 6: l.351 missing coma:

"DOTA1.0 ` HRCS2016"

Response: We have fixed the typo from ‘、’ to ‘,’ (on line 359).

Comment 7: Section Result - should be modified to the section Results and section Discussion

Response: As suggested by reviewer, we have added section discussion after section results (on line 384). In the section results, we provide information of datasets and the analysis of preliminary evaluation results. The further discussions such as the limitation and the bad detection results are mentioned in discussion section.

Comment 8: Negative results should be discussed

Response: Thank you for pointing this out. We have added the discussion of negative results in the discission section.

Reviewer 2 Report

 This manuscript proposes an oriented object detector based on YOLOv4 for oriented vehicle detection without the problem of boundary discontinuous caused by angular periodicity or corner ordering. The author presents a promising method to directly mark the objects’ heads in the field of oriented vehicle detection. However, there are some problems with the technical clarity and technical depth of the manuscript. The reviewer has also a major concerns on the accuracy improvement: is it contributed by the proposed method or YOLOv4 model itself, as there lacks a fair comparison. At the same time, there are some English writing issues and careless issues in the manuscript.

(1) The manuscript has some technical clarity problems:

The literature review and motivation is not good enough. In this manuscript, the author stress that the proposed oriented object detector doesn’t have the problem of boundary discontinuous. However, the author does not introduce the boundary discontinuous problem itself: What is the boundary discontinuous problem? What are the disadvantages of the boundary discontinuous problem? How the other works mentioned in the Introduction part fix this problem and what’s their methods’ drawbacks? Please add a brief introduction of the boundary discontinuous problem in the revised version.

In this manuscript, there are many words or expressions that are not defined or described clearly:

i. The caption of Figure 1. and the method names of (a), (b) and (c) in Figure 1. should be changed to be more formal and more accurate.

ii. In the caption of Figure 3., the “feature extraction and feature fusion parts”, the “two additional channels” and the “(C+7)A dimensions vector” should be marked in the figure or described precisely.

iii. There are many OBBs based on different methods mentioned in the manuscript, such as, “5-parameter OBB model” in line 163, “the general OBB” in line 216, etc. This will make readers confused. Please define the method name of OBBs more accurately and precisely in the revised version.

iv. In line 215, “complete angular information” is not defined or described before. It will make readers confused whether the complete angular information is the ’s range being . Please clarify in the revised paper.

v. In Figure 3, the size information of each layer’s feature maps should be marked clearly as it’s the basic and critical information of the proposed oriented object detector algorithm.

vi. In Table 1., it would be better for readers’ better view if the best AP value of each class and the best mAP value be highlighted in a different color.

       Please check through the manuscript and fix similar issues.

(2) There are also some issues in technical depth:

In this manuscript, the author stress that the proposed front point offset regression and IoU correction factor methods outperforms other methods in terms of detection speed and accuracy. However, as shown in Table 1. and Table 2., the proposed oriented object detector’s backbone is CSPDarknet53, while most of other  works’ backbones in the table are ResNet. This will make readers confused whether it is the proposed regression and correction factor methods or the CSPDarknet53(YOLO v4) itself that contributes to the faster detection speed and higher detection accuracy. The author should make an experiment on YOLO v4 with θ (angle of the bounding box) regression for oriented vehicle detection and make a deep discussion on it to show the effectiveness of the proposed regression and correction methods.

In section 3.5, part 3 (Results), the author only demonstrates the proposed oriented object detector’s detection results on their own dataset without any comparison with the results of the SOTA works transferred to the author’s dataset. In this way, section 3.5 and Table 3. seems to be not convincing and a little bit meaningless. The author should transfer the SOTA works to the author’s dataset and compare the experiment results between the proposed oriented object detector and the SOTA works to make the results more persuasive and comprehensive.  

Table 2. shows that the mAP of the proposed method is not good enough when compared with the SOTA works, and the discussion related to the Table 2 is not comprehensive and convincing enough. The author should provide a insight into the reason why the mAP of the proposed oriented object detector trained on DOTA1.0 with 15 classes is higher than trained on HRSC2016 with only 1 class.

How the FPS is measured should be described in the manuscript. If the comparison results are based on different hardware platforms, the comparison may be unfair.

Generally, there are many published good algorithm models that are open sources. It is recommended to test your own dataset with others' typical models.

(3) There are also some English writing issues and careless issues:

In paragraph 2, part 1 (Introduction), the ‘…,’ is redundant.

In line 105, the ‘roi’ should be corrected to ‘ROI’.

In line 200, “the proposed” should be changed to into a more specific and accurate expression, e.g., “the proposed method”.

In the backbone part, Figure 3, the figure and the text are overlapped. Please improve the quality of the Figure 3.

In the illustration note of Figure 3, the ‘(C+7)xA’ should be ‘(C+7)A’.

In equation (3), ‘N’ should be ‘’.

In equation (6), what is the relationship between  and ?  should be corrected to  if I’m not mistaken.

In Table 1 and 2, what does the ‘- ‘and blank mean? If they have different meaning, please add a note right under the tables.

In line 249, whether it is appropriate to mark on the title.

In line 351, ‘’ should be corrected.

In the illustration note of Figure 7, the word’s color should be changed to black.

Author Response

Thank you very much for your careful review. We have cautiously corrected typos in the manuscript and more detailed description has been added. We hope it will allay reviewer’s concerns. Below is the point-by-point response to the reviewer’s comments.

Comment 1: The literature review and motivation is not good enough. In this manuscript, the author stress that the proposed oriented object detector doesn’t have the problem of boundary discontinuous. However, the author does not introduce the boundary discontinuous problem itself: What is the boundary discontinuous problem? What are the disadvantages of the boundary discontinuous problem? How the other works mentioned in the Introduction part fix this problem and what’s their methods’ drawbacks? Please add a brief introduction of the boundary discontinuous problem in the revised version.

Response: We apologize for the inconvenience caused by the inadequate structure. A brief introduction to the boundary discontinuity problem has been added (line 50). It is a situation caused by corner miss ordering or angle periodicity while regressing the loss. It makes loss increasing instantly if the model encounters a boundary situation during training. As a result, the methods have to employ particular, often complex, techniques to mitigate this issue. The drawbacks of the other methods are mentioned (on line 98-127) and divided into three types. The first type is to regress the angle value directly which will face discontinuous problem in the boundary case due to the angle periodicity. The second type is to regress the four corner coordinates of the OBB which may not form a true rectangular shape. Although there is no angle periodicity situation, there is still a boundary discontinuous problem caused by corner ordering during training. The third type is to convert regression-based method to classification method. It avoids the boundary discontinuous problem but loses the precision caused by angle discretization and also increases the difficulty of tuning the network caused by importing the new hyperparameter window function. We also add a sentence to illustrate the disadvantages of the boundary discontinuous problem (line 50)

Comment 2: The caption of Figure 1. and the method names of (a), (b) and (c) in Figure 1. should be changed to be more formal and more accurate.

Response: Many thanks for your reminder, we have rewritten the caption of Figure 1 (line 162)

Comment 3: In the caption of Figure 3., the “feature extraction and feature fusion parts”, the “two additional channels” and the “(C+7)A dimensions vector” should be marked in the figure or described precisely.

Response: In addition to the two additional channels highlighted in red in Figure 3 (line 231), we also describe the “feature extraction and feature fusion parts”, the “two additional channels” and the “(C+7)A dimensions vector” in more detail in the caption (line 233) and section2.3 (line 193).

Comment 4: There are many OBBs based on different methods mentioned in the manuscript, such as, “5-parameter OBB model” in line 163, “the general OBB” in line 216, etc. This will make readers confused. Please define the method name of OBBs more accurately and precisely in the revised version.

Response: Thanks for reviewer’s suggestion, we have unified the names of all OBBs, and the description of "the general OBB" on line 216 has also been removed (line 224)

Comment 5: In line 215, “complete angular information” is not defined or described before. It will make readers confused whether the complete angular information is the ’s range being . Please clarify in the revised paper.

Response: Thank the reviewer for pointing this out. We have removed the words "complete angular information" and replaced them with a more detailed description (line 224).

Comment 6: In Figure 3, the size information of each layer’s feature maps should be marked clearly as it’s the basic and critical information of the proposed oriented object detector algorithm.

Response: Thanks for reviewer’s suggestion, we have added the size information of each layer’s feature maps in figure 3 (on line 231).

Comment 7: In Table 1., it would be better for readers’ better view if the best AP value of each class and the best mAP value be highlighted in a different color.

Response: We have highlighted the best mAP value of each column with red color in table 1 (on line 303).

Comment 8: In this manuscript, the author stress that the proposed front point offset regression and IoU correction factor methods outperforms other methods in terms of detection speed and accuracy. However, as shown in Table 1. and Table 2., the proposed oriented object detector’s backbone is CSPDarknet53, while most of other  works’ backbones in the table are ResNet. This will make readers confused whether it is the proposed regression and correction factor methods or the CSPDarknet53(YOLO v4) itself that contributes to the faster detection speed and higher detection accuracy. The author should make an experiment on YOLO v4 with θ (angle of the bounding box) regression for oriented vehicle detection and make a deep discussion on it to show the effectiveness of the proposed regression and correction methods.

Response: Many thanks for your suggestion. The response to this suggestion will be divided into two parts as follows.

  1. Backbone comparison

The reviewer's question is indeed the question of many researchers’, but how to compare the accuracy and speed of backbones in a completely equal way is indeed a difficult problem. Under different hardware, different framework (PyTorch, Tensorflow), different programming methods, it is hard to have a benchmark to compare. We provide the following information for reviewer’s reference. According to the experimental results from WongKinYiu, the author of yolov4, the top1 accuracy of CSPDarknet53 in ImageNet is 0.6% lower than that of ResNet152 and 1.4% higher than that of ResNet50, and there is no much difference between CSPDarknet53 and ResNet50/152 in the top5 accuracy. Although there is no actual comparison between CSPDarknet53 and ResNet101, it can be generally inferred that the feature extraction ability of CSPDarknet53 should be between ResNet101 and ResNet152. As for the inference speed of the backbone, it is also affected by many factors. Although the number of parameters and BFLOPs of the model do not directly reflect the speed of model inference, they can still be used as a reference. The number of parameters and BFLOPs of CSPDarknet53 are 27.61M and 13.07 respectively. The number of parameters and BFLOPs of ResNet50/152 are 22.73M/60.2M and 9.74/22.6 respectively. From the above information, we consider that CSPDarknet53 and ResNet101 can be discussed together.

  1. Experiment on YOLO v4 with θ

For experiments with θ regression, we worked on it in the very early stages of the method, but the results were so poor that we simply ruled it out at that time.

Comment 9: In section 3.5, part 3 (Results), the author only demonstrates the proposed oriented object detector’s detection results on their own dataset without any comparison with the results of the SOTA works transferred to the author’s dataset. In this way, section 3.5 and Table 3. seems to be not convincing and a little bit meaningless. The author should transfer the SOTA works to the author’s dataset and compare the experiment results between the proposed oriented object detector and the SOTA works to make the results more persuasive and comprehensive.

Response: Thank you for your suggestion. Our dataset mainly provides an application scenario of the propose method, highlighting the environment in which our model is suitable for operation. Our model is suitable for detecting the targets without large aspect ratio, and has a significant front side. Thus, the vehicle is a good detection target, so we do the test on our own dataset, which comes from an ongoing real project on traffic safety.

Comment 10: Table 2. shows that the mAP of the proposed method is not good enough when compared with the SOTA works, and the discussion related to the Table 2 is not comprehensive and convincing enough. The author should provide a insight into the reason why the mAP of the proposed oriented object detector trained on DOTA1.0 with 15 classes is higher than trained on HRSC2016 with only 1 class.

Response: Since the training data of DOTA has nearly 30,000 patches from nearly 3,000 original images, the training data of HRSC2016 has only 436 images and each image has only few object instances. The difference in the data makes the mAP on DOTA1.0 higher than that on HRSC2016.

Comment 11: How the FPS is measured should be described in the manuscript. If the comparison results are based on different hardware platforms, the comparison may be unfair.

Response: As the reviewer concerned, the platforms used by each method are different. However, we compare the first two methods with the fastest inference speed in DOTA1.0, Gliding Vertex and SAR. Gliding Vertex uses NVIDIA Titan Xp and SAR uses NVIDIA Tesla V100 which is the same as ours, and our method still gets higher FPS than both of the methods.

Comment 12: Generally, there are many published good algorithm models that are open sources. It is recommended to test your own dataset with others' typical models.

Response: Thank you for your suggestion. Our dataset mainly provides an application scenario of the propose method, highlighting the environment in which our model is suitable for operation. Since our dataset comes from an ongoing traffic safety project, we will release it in the future.

Comment 13: In paragraph 2, part 1 (Introduction), the ‘…,’ is redundant.

Response: We have fixed the typo (line 42).

Comment 14: In line 105, the ‘roi’ should be corrected to ‘ROI’.

Response: We have fixed the typo (line 106).

Comment 15: In line 200, “the proposed” should be changed to into a more specific and accurate expression, e.g., “the proposed method”.

Response: We have changed “the proposed” to “the proposed method” (line 209).

Comment 16: In the backbone part, Figure 3, the figure and the text are overlapped. Please improve the quality of the Figure 3.

Response: We apologize for the low image quality. We have re-drawn Figure 3 (line 233)

Comment 17: In the illustration note of Figure 3, the ‘(C+7)xA’ should be ‘(C+7)A’.

Response: We have fixed the typo (line 236).

Comment 18: In equation (3), ‘N’ should be ‘’.

Response: Sorry, we are not sure what the reviewer's question is. We noticed that there is no description of ‘N’ in the content. We have added the meaning of ‘N’ (line 243)

Comment 19: In equation (6), what is the relationship between  and ?  should be corrected to  if I’m not mistaken.

Response: Sorry, we are not sure what the reviewer's question is.

Comment 20: In Table 1 and 2, what does the ‘- ‘and blank mean? If they have different meaning, please add a note right under the tables.

Response: We apologize for confusing the reviewer. ‘-‘ and blank have the same meaning. We have renewed table 1 and 2 (line 303-304 and 351-352).

Comment 21: In line 249, whether it is appropriate to mark on the title.

Response: Sorry, we are not sure what the reviewer's question is. We have marked “DOTA”, ”HRSC2016”, “Our dataset” to section title “3.1.1. DOTA”, ”3.1.2. HRSC2016”, “3.1.3. Our dataset” (line 264,276,282).

Comment 22: In line 351, ‘、’ should be corrected.

Response: We have fixed the typo from ‘、’ to ‘,’ (line 359).

Comment 23: In the illustration note of Figure 7, the word’s color should be changed to black.

Response: Thank you. We have changed the word’s color to black (line 382).

Reviewer 3 Report

- "The experimental results on two public datasets, DOTA and HRSC2016, demonstrate that the proposed method significantly outperforms other methods in terms of detection speed while maintaining a high accuracy". The authors can mention the percentage of improvement in terms of accuracy in the abstract.

- "In order to obtain the position and orientation of objects while eliminating background interference. " The sentence ends here. But it should not end here. Please check the entire manuscript for grammatical errors and typos thoroughly and polish the English language used.

- What are the main motivations and key contributions of this study?

- Summarize the recent works in a table that depicts the state of art methodologies used for object detection through UAVs, their limitations.

- Some of the recent works such as the following can be discussed

AoI Optimization for UAV-aided MEC Networks under Channel Access Attacks: A Game Theoretic Viewpoint, A drone-based data management and optimization using metaheuristic algorithms and blockchain smart contracts in a secure fog environment.

- A detailed analysis on the results obtained that include the inferences of the authors on why and how the proposed method performed better has to be presented.

- What are the threats to validity of the proposed approach?

- Discuss about the future enhancements of this study.

Author Response

Thank you very much for your careful review. More detailed description has been added in the manuscript. We hope it will allay reviewer’s concerns. Below is the point-by-point response to the reviewer’s comments.

Comment 1: "The experimental results on two public datasets, DOTA and HRSC2016, demonstrate that the proposed method significantly outperforms other methods in terms of detection speed while maintaining a high accuracy". The authors can mention the percentage of improvement in terms of accuracy in the abstract.

Response: Thanks for the reviewer’s suggestion. We have mentioned the performance in the abstract (line 18).

Comment 2: "In order to obtain the position and orientation of objects while eliminating background interference. " The sentence ends here. But it should not end here. Please check the entire manuscript for grammatical errors and typos thoroughly and polish the English language used.

Response: Many thanks for your reminder, we have completed the sentence (line 71) and recheck the entire manuscript for grammatical errors and typos.

Comment 3: What are the main motivations and key contributions of this study?

Response: The main motivation of the proposed work is to develop an efficient vehicle detector for analyzing traffic safety at intersections. There three key contributions of our work (line 210): (1) Most previous methods cannot distinguish between the front and the rear of an object, which is an important information for vehicles. (2) The regression of angles is prone to the problem of boundary discontinuity. The proposed method regresses the coordinates of the front point to avoid the above problem. (3) Due to the boundary discontinuity problem, many previous studies added multiple anchors with different angles to help the model converge, which also increases the computational cost. Our method maintains the overall performance of the model without increasing the number of anchors.

Comment 4: Summarize the recent works in a table that depicts the state of art methodologies used for object detection through UAVs, their limitations.

Response: Thanks for reviewer’s suggestion. In this paper, we mainly concentrate on object localization with OBB on aerial images. the advantages and limitations of OBB is highly related to its application, so it is difficult to make a unified summary of these works.

Comment 5: Some of the recent works such as the following can be discussed

AoI Optimization for UAV-aided MEC Networks under Channel Access Attacks: A Game Theoretic Viewpoint,

A drone-based data management and optimization using metaheuristic algorithms and blockchain smart contracts in a secure fog environment.

Response: Thanks for reviewer’s opinion. However, these two articles seem to be less relevant to our work.

- Comment 6: A detailed analysis on the results obtained that include the inferences of the authors on why and how the proposed method performed better has to be presented.

Response: Thanks for reviewer’s opinion. The inferences about why and how the proposed approach performs better are mentioned in section 2.3 (line 207). We have also added a discussion section to show the limitations of the proposed work (line 384).

Comment 7: What are the threats to validity of the proposed approach?

Response: We have added the threats in discussion section (on line 384).

Comment 8: Discuss about the future enhancements of this study.

Response: Thanks for reviewer’s suggestion. We have added the future enhancements in discussion section (on line 384).

Reviewer 4 Report

- What was the number of anchors used ?

- What were the feature extraction layers?

- What was the positive overlap threshold used in the results? You mention 10% in Figure 4, which is very low.

- Kindly provide the precision-recall curves, ROC, and AUC.

- The conclusion need to be expanded.

- The results would be more meaningful if variations in the overlap threshold were tested. 

- The table of abbreviations is missing

- Arrange references in citation order and combine multiple references in the same square brackets (e.g., [1,2]) and be consistent (e.g., line 27 vs 116)

- Typo on line  59, pooing --> pooling

- The dataset description should be in the materials and methods section

- Section headings, only capitalize the first word.

- Restrict Table 1 to the most recent results so that the comparison can be appreciated. 

- Section headings need spacing (3.3,3.4)

Author Response

Thank you very much for your careful review. We have cautiously corrected typos in the manuscript and more detailed description has been added. We hope it will allay reviewer’s concerns. Below is the point-by-point response to the reviewer’s comments.

Comment 1: What was the number of anchors used ?

Response: There are three detection layers (head) in the proposed architecture. For each layer, we have three anchors to detect object in different aspect ratios and sizes. In total, each layer will have nine anchors in our model.

Comment 2: What were the feature extraction layers?

Response: As our network architecture is based on YOLOv4, the whole backbone, CSPDarknet53, is our feature extraction layers.

Comment 3: What was the positive overlap threshold used in the results? You mention 10% in Figure 4, which is very low.

Response: There is a slight difference between HBB and OBB in the strategy of setting the overlap threshold. Since HBB has no rotation property, it visually feels that the bounding box contains the object even if the overlap between predicted object area and actual object area is not very high. Therefore, the overlap threshold of HBB can usually be set to a higher value. On the other hand, OBB has rotation property, which makes it easy to get attention in the case of slight displacement. This is more likely to occur in the object with larger aspect ratio, thus the overlap threshold of OBB will be set lower.

Comment 4: Kindly provide the precision-recall curves, ROC, and AUC.

Response: We apologize for this. Due to the limited response time, we were unable to provide the precision-recall curves, ROC, and AUC.

Comment 5: The conclusion need to be expanded

Response: Thanks for reviewer’s opinion. We have expanded it.

Comment 6: The results would be more meaningful if variations in the overlap threshold were tested.

Response: Thanks for reviewer’s suggestion. We have tested different overlap threshold in our dataset. We got best mAP at overlap threshold 0.1 and got lower mAP at overlap threshold 0.5, 0.3, and 0.05. The difference between the best and worst mAPs was less than 0.001.

Comment 7: The table of abbreviations is missing

Response: Sorry, we are not sure what the reviewer's question is. We guess that the reviewer is referring to the abbreviations of the classes mentioned in Table 1. We have mentioned them in section 3.1.1 (on line 264-282).

Comment 8: Arrange references in citation order and combine multiple references in the same square brackets (e.g., [1,2]) and be consistent (e.g., line 27 vs 116)

Response: Many thanks. We have combined multiple references in the same square brackets.

Comment 9: Typo on line  59, pooing --> pooling

Response: We have fixed the typo (line 61).

Comment 10: The dataset description should be in the materials and methods section

Response: Thanks for reviewer’s suggestion. We have changed the section 3 from title “Results” to “Datasets and results” (line 258).

Comment 11: Section headings, only capitalize the first word.

Response: We have fixed the typos.

Comment 12: Restrict Table 1 to the most recent results so that the comparison can be appreciated.

Response: Thanks for reviewer’s suggestion. We have restricted table 1 to the most recent results (line 303).

Comment 13: Section headings need spacing (3.3,3.4)

Response: We have fixed the typos.

Reviewer 5 Report

This research work discussed the important problem of object detection. Though it is a widely worked out problem in the research community, still it is important and enhancement is always better. The overall paper is written nicely but there are some serious concerns of mine that needed to be addressed before this article may be considered for publication. The observations are as follows;

1.      Authors need to mention the pinpointed research objectives/ contribution for this work at the end of the introduction section.

2.      Why authors used YOLO4? Any specific reason why the latest versions like YOLO6 and even YOLO7 are now available, but not used? I suggest that you do experimentation at least with YOLO6 and probably you may get much better results. Researchers already used YOLO5 with OBB.

3.      Primary aim is to enhance accuracy or detection speed?

4.     As far as results are concerned, the results of state-of-the-art existing algorithms are better than the proposed work in terms of accuracy (which is more critical) as compared to detection speed and its evident from the comparison as well that very few works have calculated the detection speed (FPS). In that case, I wish to see wide improvement in the proposed work as compared to the existing techniques.

5. There are some typo errors, need to go through the article to fix those.

Author Response

Thank you very much for your careful review. We have cautiously corrected typos in the manuscript and more detailed description has been added. We hope it will allay reviewer’s concerns. Below is the point-by-point response to the reviewer’s comments.

Comment 1: Authors need to mention the pinpointed research objectives/ contribution for this work at the end of the introduction section.

Response: Thanks for your suggestion. We have added a paragraph for our objective/contribution at the end of the introduction section (line 128).

Comment 2: Why authors used YOLO4? Any specific reason why the latest versions like YOLO6 and even YOLO7 are now available, but not used? I suggest that you do experimentation at least with YOLO6 and probably you may get much better results. Researchers already used YOLO5 with OBB.

Response: YOLOv6 and YOLOv7 had not been proposed when our experiment was started. And why we did not choose YOLOv5 as based model is because it has no official published paper up to the present and the public experimental results are also similar to YOLOv4. Therefore, we choose YOLOv4 as our embedment architecture.

Comment 3: Primary aim is to enhance accuracy or detection speed?

Response: Our proposed approach focuses on locating objects with OBB and obtaining front point information while balancing the trade-off between accuracy and speed. We also hope to keep improving accuracy and speed with future upgrades of the YOLO series.

Comment 4: As far as results are concerned, the results of state-of-the-art existing algorithms are better than the proposed work in terms of accuracy (which is more critical) as compared to detection speed and its evident from the comparison as well that very few works have calculated the detection speed (FPS). In that case, I wish to see wide improvement in the proposed work as compared to the existing techniques.

Response: Thanks for the reviewer’s suggestion. It is true that there are many methods with better accuracy than ours in table1. However, our method is not only faster than others, but also achieves a good balance between accuracy and speed. Moreover, the MAP of vehicle detection of proposed method is also the best, and we can provide front side information of the object, which can be used for the traffic analysis. Finally, the accuracy and speed can be further improved with future upgrades of the YOLO series.

Comment 5: There are some typo errors, need to go through the article to fix those.

Response: Thanks for reviewer’s careful review. We have gone through the article and fixed the typos.

Round 2

Reviewer 2 Report

Although the authors have addressed our comments in this revised paper, there are still some minor issues that need to be solved before it can be accepted.

(1) In aspect of technical clarity, there are still some details to be improved:

In Figure 3., please check the overlap between “H/8 x W/8” and the input picture in “Backbone” part. At the same time, please add information of the layers’ widths and heights in “Neck” and “Head” part. Please update the figure accordingly as we suggested in the last review.

In equation (11), what is the relationship between  and ?  should be corrected to  if I’m not mistaken.

The discussion of backbone comparison in the response of comment 8 should be added in the revised manuscript for readers’ better understanding of the proposed oriented object detector.

The platforms used for testing the proposed OBB method should be added in the revised manuscript to make the experiment results more convincing and fairer.

(2) Some issues in technical depth should also be treated seriously:

The authors mentioned that the lower mAP of HRSC2016 compared with DOTA1.0 is the result of less training data. However, this comment is not convincing in terms of training CNN algorithm. Though the total numbers of the training data in two datasets are not similar, the total numbers of detection targets in two datasets may be similar, as there exists several detection targets in one image. Please provide the number of detection targets (ships) used for training in DOTA1.0 and HRSC2016 to make a deeper discussion in the revised manuscript.

(3) English writing issues and careless issues need to be corrected:

In the last columns of Table 1. and Table 2., please unify “-”and blank.

Please remove the reference mark in “3.1.1DOTA [31]” and “3.1.2. HRSC2016 [32]” (line 283, line 296).

Please proofread the paper carefully.

Author Response

We appreciate editor and reviewers for your precious time in reviewing our paper and providing valuable comments. We have carefully considered the comments and tried our best to address them. We hope the manuscript after careful revisions meet your high standards. Below we provide the point-by-point responses to the reviewers’ comments.

Comment 1: In Figure 3., please check the overlap between “H/8 x W/8” and the input picture in “Backbone” part. At the same time, please add information of the layers’ widths and heights in “Neck” and “Head” part. Please update the figure accordingly as we suggested in the last review.

 Response: Many thanks for your reminder. The overlap between “H/8 x W/8” and the input picture in “Backbone” part occurs in the pdf file. We have corrected the figure and added the size information in Neck and Head part (line 229).

Comment 2: In equation (11), what is the relationship between  and ?  should be corrected to  if I’m not mistaken.

 Response: Sorry, the question we saw is an incomplete sentence like “In equation (11), what is the relationship between  and ?  should be corrected to  if I’m not mistaken.”

Thus, we are not sure what the question is, but we've rewritten a paragraph (line 249-255) that we hope will clear up your confusion.

Comment 3: The discussion of backbone comparison in the response of comment 8 should be added in the revised manuscript for readers’ better understanding of the proposed oriented object detector.

 Response:

Thanks reviewer’s suggestion. CSPDarknet53 and ResNet are both well-known and popular backbones used for object detection. Although we spent a lot of space in our previous revision for explaining why there is little difference between the two, we are afraid that such a discussion might take away from the focus of this paper. We regret that we will not go into the issue in depth in the article.

Comment 4: The platforms used for testing the proposed OBB method should be added in the revised manuscript to make the experiment results more convincing and fairer.

Response: Thanks for reviewer’s suggestion. The platforms we used is introduced on line 257.

Comment 5: The authors mentioned that the lower mAP of HRSC2016 compared with DOTA1.0 is the result of less training data. However, this comment is not convincing in terms of training CNN algorithm. Though the total numbers of the training data in two datasets are not similar, the total numbers of detection targets in two datasets may be similar, as there exists several detection targets in one image. Please provide the number of detection targets (ships) used for training in DOTA1.0 and HRSC2016 to make a deeper discussion in the revised manuscript.

Response: Thanks for reviewer’s suggestion. We have added a small paragraph to emphasize the size difference between the DOTA1.0 and HRSC2016 (line 278-282). In DOTA1.0 the number of ships used for training is around tens of thousands. Most of the images contain dozens to s of targets. On the other hand, HRSC2016 has only 3000 targets in all. Each image contains less than ten objects.

Comment 6: In the last columns of Table 1. and Table 2., please unify “-”and blank.

Response: We have unified “-” and blank of table1 and tabel2 in previous revision (line 301, 349).

Comment 7: Please remove the reference mark in “3.1.1DOTA [31]” and “3.1.2. HRSC2016 [32]” (line 283, line 296).

Response: Many thanks for your reminder. We have removed reference mark from subtitle to the context (line 263, 275).

Comment 8: Please proofread the paper carefully.

Response: Thanks for reviewer’s suggestion. We have carefully re-checked the whole paper and correct some grammatical errors.

Reviewer 3 Report

Most of the comments are addressed. The article can be accepted for publication.

Author Response

We appreciate the editor and the reviewers for your precious time in reviewing our paper and providing valuable comments.

Reviewer 4 Report

The authors addressed most of my comments. 

Author Response

(The authors gave the same response as above.)

Reviewer 5 Report

Accepted in this form.

Author Response

(The authors gave the same response as above.)
